# Influence of Crosslinking Extent on Free Volumes of Silicone Rubber and Water Diffusion after Corona Discharge

**DOI:** 10.3390/ma15196833

**Published:** 2022-10-01

**Authors:** Yue Yang, Zheng Wang, Xiangyang Peng, Zhen Huang, Pengfei Fang

**Affiliations:** 1School of Physics and Technology, Wuhan University, Wuhan 430072, China; 2Guangdong Key Laboratory of Electric Power Equipment Reliability, Electric Power Research Institute of Guangdong Power Grid Co., Ltd., Guangzhou 510080, China

**Keywords:** silicone rubber, crosslinking level, corona discharge, electrochemical impedance, water diffusion

## Abstract

Silicone rubber is widely used as an insulating material. In this article, silicone rubber samples were prepared by varying the content of crosslinker (2,5-bis(tert-butyl-peroxy)-2,5-dimethylhexane, DBPMH), and the free volume holes in the samples were investigated by means of positron annihilation lifetime spectroscopy (PALS) measurement. The surface chemical structure, surface micromorphology and water diffusion of the samples after corona discharge treatment were studied by FTIR, SEM and EIS measurements, respectively. As the crosslinker weight ratio increased from 0.2 wt.% to 1.5 wt.%, the mean free volume hole size first decreased and then remained unchanged. However, the concentration of free volume holes did not vary as the crosslinker weight ratio increased. SEM morphologies show that surface cracks were produced on samples having high crosslinking levels after corona treatment. The water diffusion coefficient of samples after corona treatment increased from 3.13 × 10^−10^ cm^2^ s^−1^ to 17.68 × 10^−10^ cm^2^ s^−1^ in the initial immersion period, as the crosslinker weight ratio increased from 0.2 wt.% to 3.0 wt.%. The results indicated that deterioration of samples with high crosslinking levels were more serious and water repellency more easily lost. The corona resistance ability of low crosslinking level silicone rubber stems from internal low molecular weight molecules.

## 1. Introduction

Silicone rubber is extensively applied in the production of composite insulators, due to its hydrophobicity, high thermal stability, and hydrophobic recovery property [1,2]. As third-generation polymer insulators, composite insulators have been widely used in high-voltage transmission lines, because of their ability to withstand contamination and the hydrophobicity recovery property given by their silicone rubber sheaths [3]. However, being organic in nature, the silicone rubber sheath can get aged under outdoor environmental stresses which leads to deterioration or even mechanical failure [4,5,6,7,8]. Operating in strong electric fields, composite insulators are easily exposed to corona discharge triggered by edges or water droplets, leading to aging and, finally, structural failure [9,10].

The structural fractures of composite insulators can be sorted into two categories: brittle fracture and decay-like fracture [11,12,13]. However, whether brittle fracture or decay-like fracture, the fracture is correlated with ingress of water into the sheath-core interface [14,15,16,17]. Long-term corona discharge exposure leads to loss of water repellency of silicone rubber material not only because of the high energy electrons, but also because of the acid gas triggered by high voltage discharges in air [18]. Tokoro et al. found that water absorption increases the dielectric constant of silicone rubber [19]. Sundfors et al. concluded that water uptake in silicone rubber membranes was very low compared to other organic membranes [20]. Gao et al. found that high aluminum hydroxide (ATH) content in silicone rubber leads to a larger amount of water being absorbed [16]. The adding of ATH particles improves the thermal conductivity and fire retardance of silicone rubber which can enhance thermal stability [21,22,23]. Wang et al. investigated the influence of corona discharge time and different inorganic filler contents on the water repellency of silicone rubber [24,25]. In conclusion, current studies have focused on the influence of inorganic fillers to the water repellency of silicone rubber.

However, composite insulator silicone rubber is a polydimethylsiloxane (PDMS)-based composite with a vinyl molar concentration of 0.8–1.6% [18]. The vinyl groups in the PDMS molecular skeleton get crosslinked during the vulcanizing process and, thus, form a three-dimensional network structure [26,27,28]. So, the vinyl crosslinking level in silicone rubber determines its molecular structure and material properties. Zhang et al. found that as the crosslinking density increases the conductivity of silicone rubber first increases and, then, decreases [29]. Warley et al. investigated dynamic modules of silicone rubber synthesized by different kinds of peroxide crosslinkers and found that benzoyl peroxide appeared to be advantageous because of its minimum Payne effect [30]. Chen et al. concluded that the phenyl silicone rubber vulcanized by polysilazane had the optimum thermal stability because of initiating depolymerization, by the random scission process of the main polymer chain [31]. Up to now, the influence of crosslinking level to the water repellency of silicone rubber has not been reported. In this study, the influence of crosslinking level on the water repellency of silicone rubber and the water diffusion process after corona discharge was investigated.

## 2. Materials and Experimental Method

### 2.1. Sample Preparation

Silicone rubber samples with different crosslinking levels were prepared by adding diverse weight ratios of crosslinker in the vulcanization process. The synthesis process was divided into several steps. The first step was mixing the PDMS matrix and fumed silica (FS) in a rubber dispersion kneader for 2–3 h in order to disperse silica evenly into the PDMS matrix. Then, silicone oil, silane coupling agent and ATH were added into the kneader for another 3 h mixing process. The second step was dispersion of the crosslinker. The silicone rubber from the previous step was first mixed on a rubber mixing mill, and then the crosslinker was added into the rubber and mixed for two minutes. The last step was vulcanization of the silicone rubber in a special mold in which 60 µm thick silicone rubber specimens could be vulcanized under a pressure of 150 MPa at 180 °C. However, another 3 mm thick mold was used for investigating the free volume in the silicone rubber sheets. 

The PDMS used in this study was methyl vinyl silicone rubber (vinyl content varied from 0.12~0.16% with relative molecular weight from 450~700 thousand). The crosslinker used was 2,5-bis(tert-butyl-peroxy)-2,5-dimethylhexane (DBPMH) which is popular in the silicone rubber production industry. The weight ratio of PDMS, ATH, FS and silicone oil in the first step was 100:100:25:16. In this study, the weight ratio of crosslinker added into the silicone rubber was 0.2%, 0.5%, 1.0%, 1.5%, 2.0%, 3.0%, respectively.

### 2.2. Corona Discharge Treatment

The silicone rubber specimens were treated to corona discharge by a homemade high voltage apparatus to simulate the operation conditions of composite insulators. A high voltage tester (MS2674, Minsheng Instrument, Nanjing, China) was used to provide high voltage output. The applied output voltage was 10 kV. A stainless-steel needle was used as the positive electrode and a steel cylinder as the negative electrode. In this high voltage apparatus, silicone rubber samples were horizontally placed on the cylinder electrode beneath the needle electrode which was vertically placed. The distance between the needle tip and the surface of the samples was 3 mm. The schematic of the high voltage apparatus is shown in Appendix A. The results of electric field strength simulation are shown in Appendix A.

### 2.3. Positron Annihilation Lifetime Spectroscopy

The correlation of free volume holes with the crosslinking level of silicone rubber was investigated by positron annihilation lifetime spectroscopy (PALS). In this study, ^22^Na sealed between two 7.5 μm thick Kapton foils was applied as the positron source. Since the penetration depth of positrons emitted by ^22^Na source in polymer is between 1~2 mm, the silicone rubber samples of different crosslinking levels prepared for PALS measurement was 3 mm thick. Two sheets of the same samples were tightly attached to both sides of the positron source to ensure all the positrons were annihilating in the samples. For measurement, a fast–fast coincident system, with a time resolution of nearly 300 ps, was used for the gamma photo signal collection. The lifetime spectra were decomposed by the PATFIT program with a fitting variance of 1.01–1.15.

### 2.4. Electrochemical Impedance Spectroscopy Measurement

The diffusion process of water permeating into the silicone rubber samples was characterized by electrochemical impedance spectroscopy (EIS) measurement (CS310, Corrtest Instrument, Wuhan, China), which is widely used in studies of coating corrosion. In this study, the 60 μm thick silicone rubber samples were placed on the surface of a piece of indium tin oxide (ITO)-coated glass. The classical three-electrode measurement system was applied in the EIS measurement, and the ITO glass served as working electrode, the Pt electrode as the counter electrode, and the saturated calomel electrode as the reference electrode, respectively. A glass tube of diameter of 12 mm, to contain the electrolyte solution (3.5 wt.% NaCl solution), was sealed on the silicone rubber samples by room temperature vulcanized silicone rubber which also fixed the silicone rubber samples on the ITO glass surface. The working area of the working electrode was the contact area of electrolyte and silicone rubber samples which was approximately 100 mm^2^. For each sample, the sinusoidal alternating voltage applied on the working electrode was 20 mV, and the applied frequency in the system ranged from 10^5^ Hz to 10^−2^ Hz. The schematic diagram of EIS measurement is shown in Appendix A.

### 2.5. Characterization

The surface chemical groups were analyzed by attenuated total reflection Fourier transform infrared (ATR-FTIR) spectroscopy (Nicolet iS10, Thermo Fisher Scientific, Waltham, MA, USA). The applied frequency range, number of scans, and resolution in FTIR measurement were 4000–650 cm^−1^, 32 and 4 cm^−1^, respectively. The crystal used in the ATR accessories was diamond. The surface micromorphology was characterized by scanning electron microscopy (SEM, Hitachi S-4800, FEI Company, Hillsboro, OR, USA). The sample surfaces were platinum-coated before SEM measurement and the applied accelerating voltage was 2 kV. For each characterization, one piece of corona treated sample with different crosslinker weight ratio was prepared.

## 3. Results and Discussion

### 3.1. Free Volume Holes in Silicone Rubber with Different Crosslinking Levels

PALS is a kind of sensitive probe for microstructure characterization in materials [32]. When a positron is emitted into substances, it may annihilate with an electron (direct annihilation), or first form a positronium (a hydrogen-like atom) with an electron and then annihilate. The positronium can be separated into two kinds: para-positronium (*p*-Ps) and ortho-positronium (*o*-Ps). In general, the mean lifetime of positron direct annihilation, *p*-Ps annihilation and *o*-Ps annihilation is 2–3 ns, 0.125 ns and 142 ns, respectively. Usually, an *o*-Ps diffuses or moves into a micropore or cavity in substances under the stress of lattice collision and then annihilates. However, sometimes the *o*-Ps in a micropore annihilates with an electron (pick-off annihilation) from the pore wall and reduces the *o*-Ps lifetime to several ns. The *o*-Ps pick-off rate is correlated to the pore dimension and electron distribution near the *o*-Ps atom, so the *o*-Ps lifetime is determined by the microstructure of substances.

For polymers, a free volume theory is applied to explain long lifetime *o*-Ps annihilation. When a positron is injected into polymers it forms an *o*-Ps with an electron after a thermalization process, and the *o*-Ps diffuses into free volume holes in molecules and, finally, annihilates or jumps into another hole. The lifetime of *o*-Ps and annihilation intensity in free volume holes are determined by the dimension and concentration of the free volume holes.

A three-lifetime decomposition method was employed in this study, in which τ_1_, τ_2_ and τ_3_ were attributed to *p*-Ps annihilation, positron direct annihilation and *o*-Ps annihilation, respectively. The lifetime of *o*-Ps of silicone rubber with diverse weight ratios of crosslinkers are shown in Figure 1. As the crosslinker weight ratio increased, the *o*-Ps lifetime in silicone rubber decreased from 3.45 ns to 3.30 ns first and then remained approximately 3.30 ns when the weight ratio exceeded 1.5%.

As mentioned earlier, the *o*-Ps lifetime is determined by the dimension of free volume holes. A square well model was applied to calculate the mean size of free-volume holes. The free volume in which an *o*-Ps is trapped is assumed to be an infinite spherical square well. The correlation between mean radius *R* of free volumes and thickness Δ*R* of electron layer outside the free volume holes with *o*-Ps annihilating rate is shown in Equation (1). The volume of the spherical free-volume holes can be calculated by Equation (2).

In general, the value of Δ*R* in polymers is 1.656 Å. So, the mean radius *R*_f_ and volume *V*_f_ of free = volume holes in silicone rubber with diverse weight ratios of crosslinkers can be calculated by Equations (1) and (2). As shown in Table 1, the radius *R*_f_ of the free-volume holes decreased from 3.92 Å to 3.82 Å as the weight ratio of crosslinker rose from 0.2 wt.% to 1.5 wt.%, and the *R*_f_ remained basically unchanged when the weight ratio exceeded 1.5 wt.%. However, the variation *o*f *o*-Ps was small as the weight ratio of crosslinker changed [33,34].
(1)$$\tau_{o-Ps}^{-1}=2\left[1-\frac{R}{R+\Delta R}+\frac{1}{2\pi }\sin\left(\frac{2\pi R}{R+\Delta R}\right)\right]$$
(2)$$V_{o-Ps}=\frac{4}{3}\pi R^{3}$$

In the vulcanization process, DBPMS decomposed to peroxy radical and reacted with vinyl groups to get the silicone molecular crosslinked structure. As the crosslinker weight ratio increased, the proportion of vinyl groups crosslinked also increased, leading to a more compact network structure. The *o*-Ps lifetime results indicated that the mean size of free volume holes in silicone rubber was determined by the crosslinking structure, and a more compact network structure meant a smaller free volume hole size. The results also indicated that the network structure tended to be saturated when the crosslinker weight ratio exceeded 1.5 wt.%. However, the variation of *o*-Ps intensity was quite small which indicated that the vinyl crosslinking process had little effect on the free volume concentration in silicone rubber.

### 3.2. Surface Chemical Structure

Surface chemical structure of silicone rubber with diverse crosslinker weight ratios before and after corona treatment was characterized by ATR-FTIR spectroscopy. According to the Beer-Lambert Law, absorbance is correlated to concentration and thickness of samples. In this study, the preparation and characterization of the samples were under the same conditions. The penetration depth of ATR is several µm, which is much smaller than the thickness of the samples (60 µm) used in this study. The absorption of attenuated infrared ray in the samples was adequate. Therefore, the intensity and peak areas of absorbance peaks in the ATR-FTIR spectrum were correlated with the concentration of chemical groups [35,36]. The intensity of characteristic peaks of silicone rubber before corona treatment were undistinguishable. The FTIR spectrum of silicone rubber with diverse cross-linker weight ratios is shown in Appendix A. Since the vinyl content of PDMS in this study was below 0.16% (mole ratio), so the intensity of the vinyl absorption peak was very low and the variation of vinyl concentration was undetectable by FTIR. 

As shown in Figure 2, the intensity of characteristic peaks of silicone rubber after corona treatment at 1260 cm^−1^, 1008 cm^−1^ and 788 cm^−1^ decreased as the crosslinker weight ratio rose from 0.2 wt.% to 1.5 wt.% The absorption peaks at 1260 cm^−1^ and 788 cm^−1^ were attributed to methyl groups, and the peak at 1008 cm^−1^ was attributed to Si-O bond. The results suggested that methyl groups on the side-chain were degraded during the corona process and the polymer main-chain became decomposed. The degrading of methyl groups and Si-O bonds became more serious as the crosslinker weight ratio increased. However, the area variation of these three absorbance peaks was small when the crosslinker weight ratio exceeded 1.5 wt.%. As for the absorbance peaks at 3250 cm^−1^, 1316 cm^−1^ and 1420 cm^−1^, the peak area variation along with the crosslinker weight ratio was opposite to the peaks ascribed to methyl groups and Si-O bonds. The absorbance peak at 3450 cm^−1^ was attributed to ATH in silicone rubber before corona treatment. However, this peak disappeared after corona treatment, and a new peak ascribed to hydroxyl vibration at 3250 cm^−1^ appeared, as shown in Figure 2. The results implied that the ATH in silicone rubber decomposed under corona discharge and water vapor was absorbed into the sample surface. The absorbance peaks at 1316 cm^−1^ and 1420 cm^−1^ were attributed to nitrogen oxides produced in the corona discharge process. As the crosslinker weight ratio increased, the peak area corresponding to nitrogen oxides became larger. These results indicated that the surface degrading of silicone rubber with higher crosslinker weight ratio was more serious than with lower crosslinker weight ratio. For silicone rubber with lower crosslinking levels, the degree of freedom of molecules’ motivation was higher. The degraded molecules at the sample surface under corona discharge could move to a deeper position through vibration or rotation. On the other hand, the inner low molecular weight molecules could diffuse to the sample surface after surface deterioration happened and absorb the energy of the corona discharge by decomposition, thus finally protecting the sample surface. However, for the high crosslinker weight ratio samples, the network structure was stable which led to a low degree of freedom of molecules’ motivation. What is more, the diffusion of inner low molecular weight molecules was limited in high crosslinking level samples due to the low movement freedom. So, the surface degradation of high crosslinker level samples was more serious than for samples of low crosslinker level. The surface chemical structure of samples of crosslinker weight ratio exceeding 1.5 wt.% were similar. This might be because the crosslinking structure became saturated when the crosslinker weight ratio exceeded 1.5 wt.%.

### 3.3. Surface Micromorphology

The surface of silicone rubber before corona discharge was quite smooth and some micro-sized particles could be observed, as shown in Appendix A. The skin-like smooth part was the organic polymer base, and the microparticles are the dispersed ATH. In addition, the surface morphology of silicone rubber samples with different crosslinker weight ratios were quite similar.

SEM images of silicone rubber with diverse crosslinker weight ratios after corona discharge are shown in Figure 3. The surface of all the samples after corona treatment were very course and there was manifest flaky matter on the surface. This flaky matter might be attributed to residue of PDMS dissociation and products of ATH decomposition. As discussed in the FTIR spectrum results, the composition of the flaky matter produced under corona discharge is complicated, and includes alumina, silica dioxide, residue of PDMS decomposition and nitrogen oxide. The SEM spectrum results also implied that manifest cracks were produced on the surface of silicone rubber samples with crosslinker weight ratio exceeding 1.5 wt.%. However, the cracks in samples with 1.5 wt.% crosslinker weight ratios were inconspicuous and no surface cracks were observed in samples with crosslinker weight ratio below 1.5 wt.%. The results indicated that the degradation of silicone rubber samples with high crosslinker weight ratio was more serious than for samples with low crosslinker weight ratio. The results of SEM spectrum were consistent with those of FTIR spectrum.

### 3.4. Water Diffusion Process

The water diffusion process in silicone rubber samples was characterized by EIS measurement. The modulus values of silicone rubber with diverse crosslinker weight ratios before corona discharge exceeded 10^9^ Ω cm^2^ at low frequencies, as shown in Figure 4. The impedance/frequency curve was nearly a straight line which was a symbol of capacitor character. The results indicated that silicone rubber virgin samples repelled water well and the NaCl solution could not diffuse into the samples, even after 50 h of immersion.

As shown in Figure 5a, the modulus of corona aged silicone rubber samples with 0.2 wt.% crosslinker weight ratio exceeded 10^9^ Ω cm^2^ at the initial period of immersion. At the same time, the phase angle values remained at a high level and dispersed at low frequencies. The reasons for this dispersion phenomena might be that the output current detected from the counter electrode was too small and even reached the detection limit of the device. After two more hours of immersion, the modulus values at low frequencies decreased to approximately 1−3 × 10^8^ Ω cm^2^ and the phase angle at low frequencies declined. However, the variation of modulus and phase angle curves was quite small as the immersion continued to 50 h. In general, a low phase angle represented resistance characteristics along with a low modulus value. As the FTIR and SEM results indicated that hydrophobic organic groups on the sample surface were degraded and hydrophilic nitrogen oxides produced during the corona treatment, the water repellent ability of the sample was partly lost. The EIS results indicated that surface degradation and delamination of the silicone rubber sample with low (0.2 wt.%) crosslinker weight ratio occurred during the corona aging process, and water could diffuse into the sample accordingly.

However, the extent of surface degradation and delamination was limited and the amount of ingress water was little. For the sample with 0.5 wt.% crosslinker weight ratio, the modulus value and phase angle at different frequencies were similar to the sample with 0.2 wt.%, as shown in Figure 5b. The results showed that the aging condition of the sample with 0.5 wt.% crosslinker weight ratio after corona treatment was similar to that of the sample with 0.2 wt.%. The modulus at low frequencies of corona aged samples with 1.0 wt.% crosslinker weight ratio exceeded 10^8^ Ω cm^2^ at the initial period of immersion (<2 h), as shown in Figure 5c. After 8 h immersion, the modulus at 0.1 Hz decreased to 5 × 10^7^ Ω cm^2^ and the phase angle was very close to 0 degree. The results indicated that the ingress of NaCl solution into the corona aged sample with 1.0 wt.% crosslinker weight ratio was more than for samples with 0.2 wt.% and 0.5 wt.%. However, it was obvious that there was only one time constant at high frequencies appearing in impedance diagrams of the sample with 1.0 wt.% cross-linker weight ratio, as shown in Figure 5c. This meant no extra relaxation process existed in the electrochemical system but the charge and discharge process at the interface between the sample surface and NaCl solution. The results indicated that there was more NaCl solution diffused into the sample during the immersion process, compared to the samples with lower crosslinker weight ratios, but did not penetrate the sample. The modulus values of corona aged sample with 1.5 wt.% crosslinker weight ratio was close to sample with 1.0 wt.%, as shown in Figure 5d. The phase angle at low frequencies remained close to −45°; however, no time constant appeared. The equivalent circuits of corona aged samples with crosslinker weight ratios of 0.2, 0.5, 1.0, 1.5 wt.% are shown in Figure 5. The equivalent circuits of these four corona aged samples after NaCl solution immersion were the same, where R_e_, C_b_ and R_b_ represent solution resistance, sample capacitance and sample resistance, respectively. As discussed above, the silicone rubber samples before corona treatment appeared very close to pure capacitances in impedance diagrams, which corresponded to C_b_ in equivalent circuits. However, the hydrophobic methyl groups were degraded and surface defects were produced under corona discharge. Thus, electrolyte solution could diffuse into the sample and give the sample resistance characteristics, which corresponded to R_b_ in equivalent circuits.

The impedance diagrams of high crosslinker weight ratio samples after corona treatment were quite different. The modulus values at 0.1 Hz decreased sharply from nearly 10^9^ Ω cm^2^ to 10^7^ Ω cm^2^ at the initial immersion period, as shown in Figure 6a. After 1.67 h immersion, the phase angle curve had a minimal value (nearly 15°) between 10 Hz and 1 Hz and increased to 60° at 0.1 Hz. The decline of phase angle from 90° to 15° was attributed to sample capacitance and the rise from 15° to 60° attributed to another electrochemical process. A Warburg impedance element was introduced to simulate the new electrochemical process, as shown in Figure 6a. The appearance of Warburg impedance pointed to a mass transfer process near the electrode which corresponded to the diffusion of electrolyte solution in this electrochemical system. As shown in Figure 3e, visible cracks were produced on the surface of high crosslinker weight ratio samples after corona discharge. It could be deduced that the production of cracks provided diffusion tunnels for the electrolyte solution. Thus, the modulus values at 0.1 Hz decreased very fast, along with the ingress of electrolyte solution at the initial immersion period. After 8 h immersion, a second time constant appeared at low frequencies close to 0.01 Hz, as shown in Figure 6a. The appearance of a second time constant meant a new relaxation process and a different equivalent circuit were introduced to simulate the impedance curves after 8 h immersion. More specifically, the Warburg element was replaced by a combination of a capacitance C_dl_ and a resistance R_ct_. In this electrochemical system, the electrolyte solution might penetrate the sample to reach the sample/ITO interface after a long time of immersion and, hence, form a solution layer. The double layer capacitance of the solution layer corresponded to C_dl_ and the charge transfer resistance of the irons in the solution under AC operating voltage corresponded to R_ct_. The impedance diagrams of corona aged sample with 3.0 wt.% crosslinker weight ratio were similar to the sample with 2.0 wt.%. The difference was the appearance of Warburg impedance and the double layer of the corona aged sample with 3.0 wt.% crosslinker weight ratio was faster than for the sample with 2.0 wt.%. The results indicated that the water diffusion rate in corona aged sample with 3.0 wt.% crosslinker weight ratio was bigger than that of the sample with 2.0 wt.%. The reason might be that there were more microcracks produced in the samples with 3.0 wt.% crosslinker weight ratio during the corona treatment, or the dimension of the microcracks was larger than in samples with 2.0 wt.%.

The evaluation of sample resistance R_b_ to the immersion time in NaCl solution of silicone rubber with diverse crosslinker weight ratios after corona discharge is shown in Figure 7a. As the immersion process was prolonged, the values of R_b_ for all the samples first rapidly decreased and then remained unchanged. The final resistance during the immersion process of low crosslinking level samples (below 1.5 wt.%) were very close and much bigger than the resistance of high crosslinking level samples (exceed 1.5 wt.%). In EIS measurement, the final sample resistance was determined by sample porosity and the value of porosity could be calculated by the value of the final sample resistance [37]. As shown in Figure 7b, the porosity of low crosslinker weight ratio (below 1.5 wt.%) samples was quite low and sample porosity increased dramatically when crosslinker weight ratio exceeded 1.5 wt.%. As mentioned earlier, the crosslinking structure tends to be saturated when crosslinker weight ratio exceeds 1.5 wt.%. The results indicated that deterioration in high crosslinking extent silicone rubber samples after corona discharge was more serious than for low crosslinking extent samples. The results also correlated with the results of FTIR and SEM measurements.

The capacitance of samples C_b_ after immersion could be calculated by equivalent circuit fitting. As shown in Figure 8, C_b_ increased very fast at the initial immersion period and then stabilized as the immersion was prolonged. Studies by Brasher and Kingsbury [38] indicate that ingress of water into polymer coatings leads to a change in the permittivity and the variation is dependent on the water uptake (ϕ) of the coating.
(3)ϕ=lg(C/tC0)lgε

In Equation (3), *C*_t_ and *C*_0_ are the capacitance at immersion time t and initial immersion, and *ε* is the permittivity of water. In general, the diffusion of water into the sample is assumed to obey Fick’s second law. Studies by Crank indicate that the amount of water permeated into a film satisfies Equation (4) [39,40]:(4)$$\frac{M_{t}}{M_{\infty }}=2\frac{\sqrt{Dt}}{L}(\frac{1}{\sqrt{\pi }}+2\sum_{n=0}^{n=\infty }{\left(-1\right)}^{n}\mathrm{ierf}c\frac{nL}{\sqrt{Dt}})$$

In Equation (4), *M_t_* and *M*_∞_ are the amount of water permeated in the sample at immersion time *t* and after fully diffusion. And *D* and *L* are the diffusion coefficient and sample thickness, respectively. For small values of *t*, Equation (4) can be reduced to:(5)$$\frac{M_{t}}{M_{\infty }}=2\frac{\sqrt{Dt}}{L\sqrt{\pi }}$$

At the initial period of immersion, the water uptake is linearly related to *t*^1/2^. So, the diffusion coefficient *D* can be calculated by Equation (5). Combining Equation (3) and Equation (5) leads to:(6)$$\frac{\mathrm{lg}(C_{t}/C_{0})}{\mathrm{lg}(C_{\infty }/C_{0})}=\frac{2\sqrt{D}}{L\sqrt{\pi }}\sqrt{t}$$

In Equation (6), *C*_0_ and *C*_∞_ are constants, so the diffusion coefficients *D* can be calculated through lg*C*_t_ and *t*^1/2^. As shown in Figure 8, the values of lg*C*_t_ are linearly related to values of *t*^1/2^, which also indicates that the diffusion of water into samples at the initial immersion period obeys Fick’s second law. The diffusion coefficient values of silicone rubber with diverse crosslinker weight ratios after corona discharge are shown in Figure 9. As the crosslinker weight ratio increased, the diffusion coefficient also increased. For high crosslink level samples (crosslinker weight ratio exceeding 1.5 wt.%), the diffusion coefficients were much bigger than for low crosslink level samples. 

Based on the above discussions, aging of silicone rubber with high crosslinker weight ratio after corona discharge is more serious than it is for low crosslinking silicone rubber. The water uptake and diffusion coefficients of high crosslinking samples after corona discharge are bigger than for low crosslinking samples. The high energy charged ions generated by high voltage alternating electric field are emitted to the sample surface leading to polymer chain breakage and molecule decomposition. Studies have shown that corona aging of silicone rubber turns the hydrophobic surface into a hydrophilic one, which accelerates the water diffusion process. The breakage of the polymer chai leads to structural damage and even to the production of microcracks (as shown in Figure 3). However, in low crosslinking level silicone rubber samples, the polymer chain has more freedom of movement. During the corona treatment process, un-crosslinked polymer chain segments near the surface can move to the sample surface which can retard the surface deterioration process and keep the surface hydrophobic to some extent. Besides, the corona discharge treatment leads to decomposition of low molecular weight molecules on the silicone rubber surface which is hydrophobic. In low crosslinking level samples, inner low molecular weight molecules can diffuse to the corona-treated surface, due to the higher molecule movement freedom. The low molecular weight molecules can absorb the corona energy by decomposition and also help to keep the surface hydrophobic.

## 4. Conclusions

In this work, different crosslinking level silicone rubber samples were prepared by controlling the content of crosslinker. Free volume holes of silicone rubber with different crosslinking levels and water transportation of silicone rubber after corona discharge treatment was systematically investigated. With the increase of crosslinker weight ratio, mean size of free volume holes first decreased from 3.92 Å to 3.82 Å (radius) and then remained unchanged (weight ratio exceeding 1.5 wt.%). However, concentration of free volume holes did not vary with crosslinking level. In high crosslinking level silicone rubber samples, decomposition of chemical groups was more serious than in low crosslinking level samples and microcracks were produced after corona discharge treatment. The water diffusion coefficient in high crosslinking lever silicone (crosslinker weight ratio exceeding 1.5 wt.%) was much bigger than in low crosslinking level silicone after corona discharge treatment. This work provides reference and guidance for the synthesis and production of composite insulator silicone rubber. The recommended crosslinker (DBPMH) weight ratio for composite insulators production is 1.5 wt.%.

## Figures and Tables

**Figure 1 materials-15-06833-f001:**
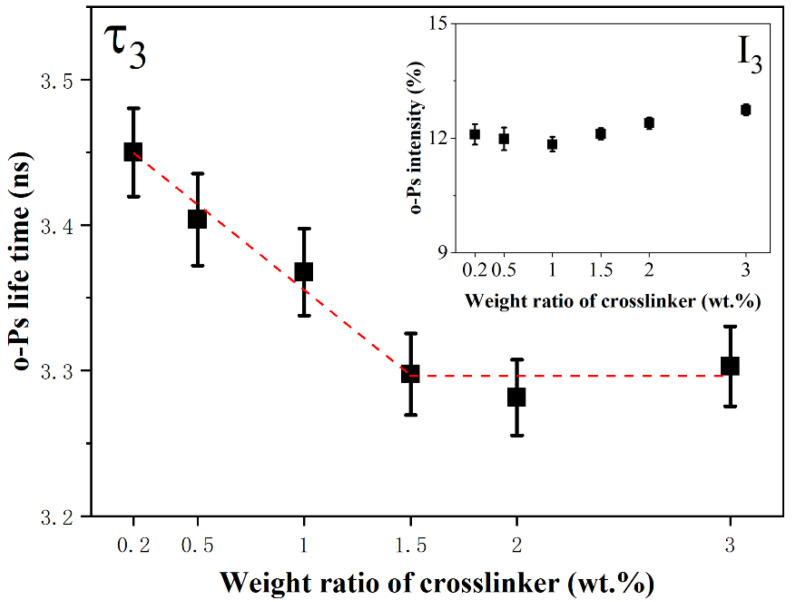
*o*-Ps lifetime a of silicone rubber with diverse weight ratio of crosslinker.

**Figure 2 materials-15-06833-f002:**
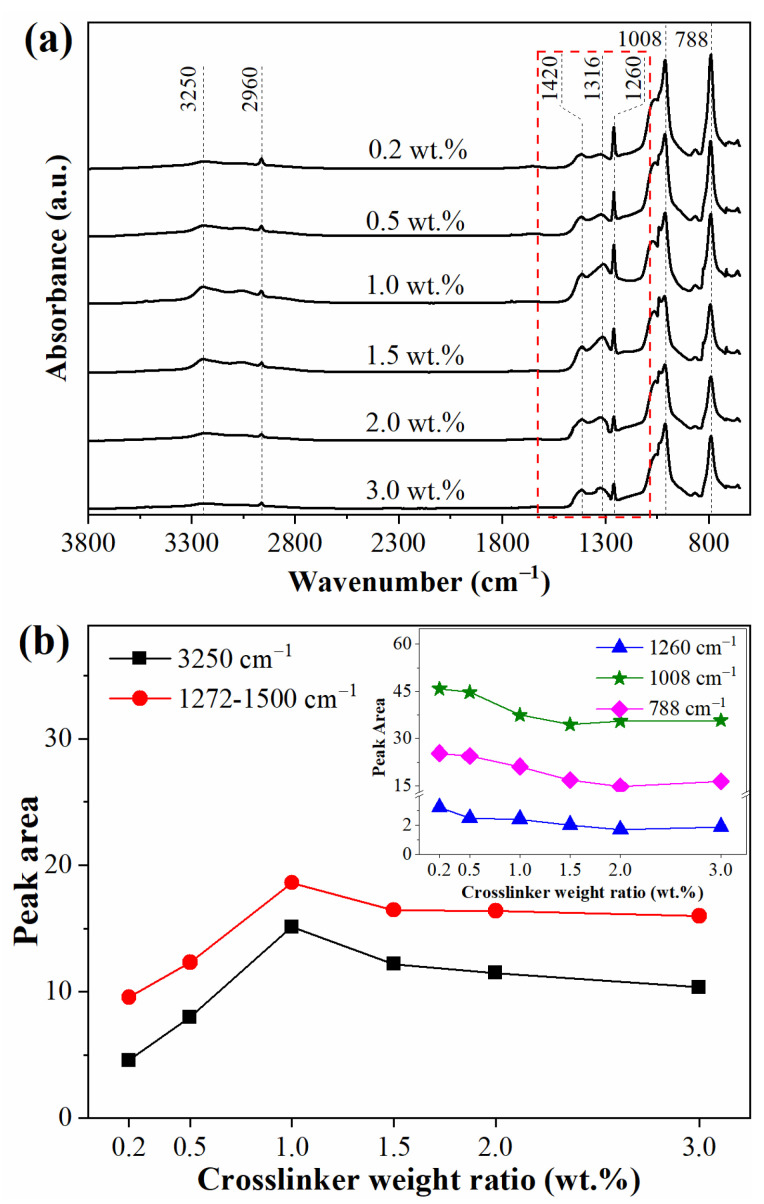
FTIR spectrum of silicone rubber with diverse crosslinker weight ratios after corona treatment (**a**); area of parts of characteristic peaks (**b**).

**Figure 3 materials-15-06833-f003:**
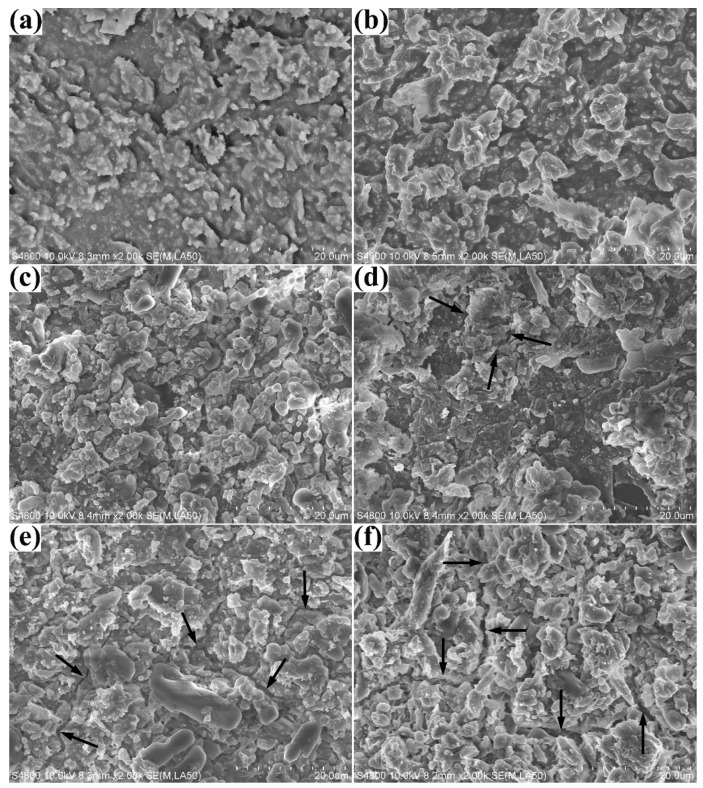
SEM images of silicone rubber with diverse crosslinker weight ratio: 0.2 wt.% (**a**), 0.5 wt.% (**b**), 1.0 wt.% (**c**), 1.5 wt.% (**d**), 2.0 wt.% (**e**) and 3.0 wt.% (**f**), respectively.

**Figure 4 materials-15-06833-f004:**
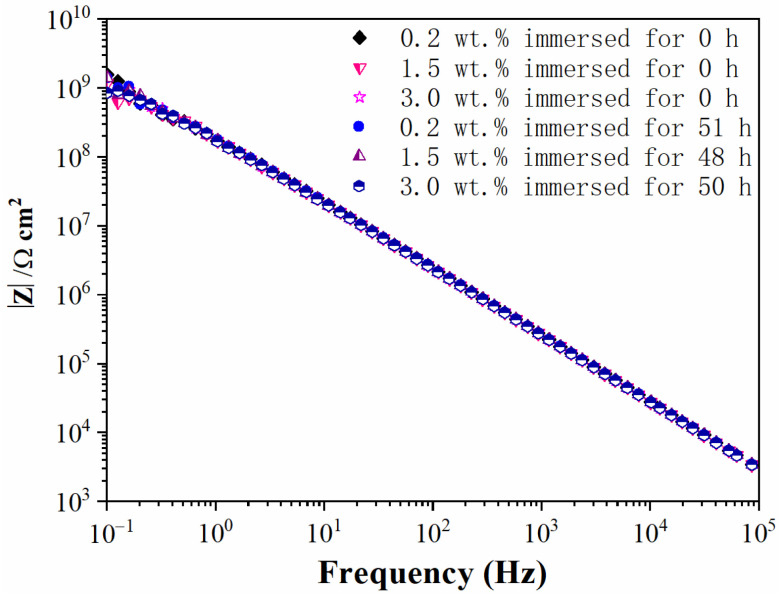
Impedance modulus values of silicone rubber samples before corona treatment.

**Figure 5 materials-15-06833-f005:**
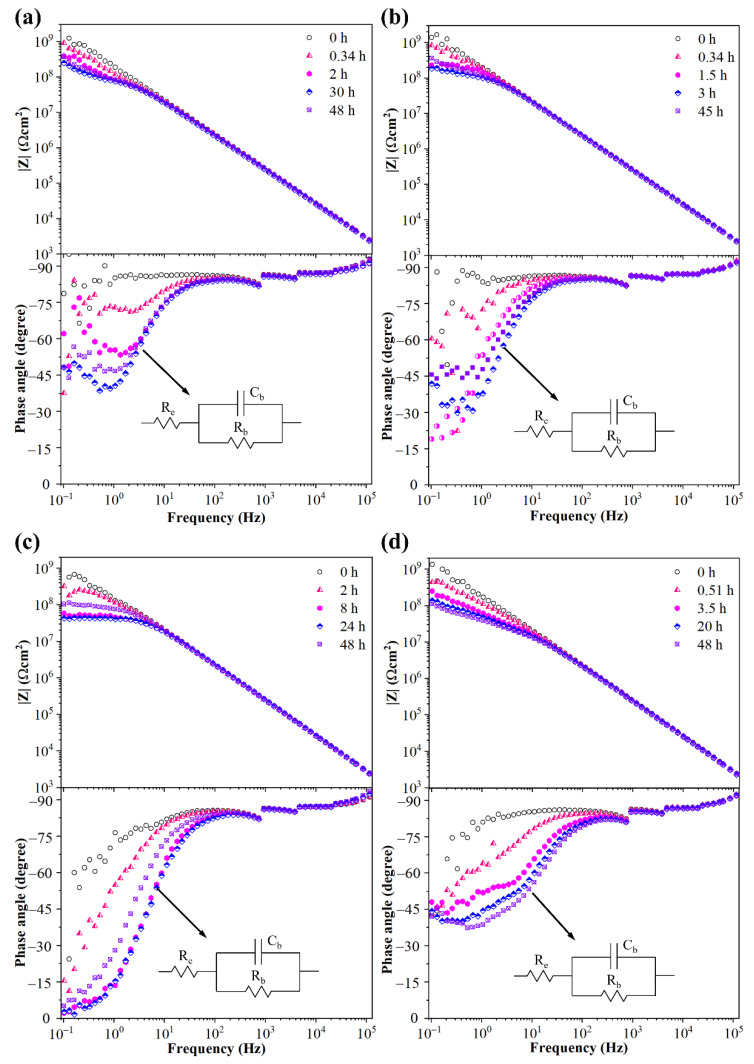
Impedance diagrams and equivalent circuits of corona aged silicone rubber samples with 0.2 wt.% (**a**), 0.5 wt.% (**b**), 1.0 wt.% (**c**) and 1.5 wt.% (**d**) crosslinker weight ratio.

**Figure 6 materials-15-06833-f006:**
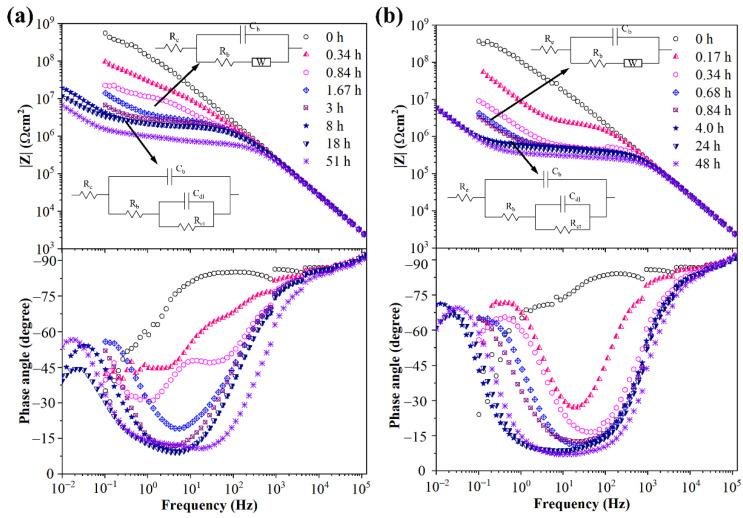
Impedance diagrams and equivalent circuits of corona aged silicone rubber with 2.0 wt.% (**a**) and 3.0 wt.% (**b**) crosslinker weight ratio.

**Figure 7 materials-15-06833-f007:**
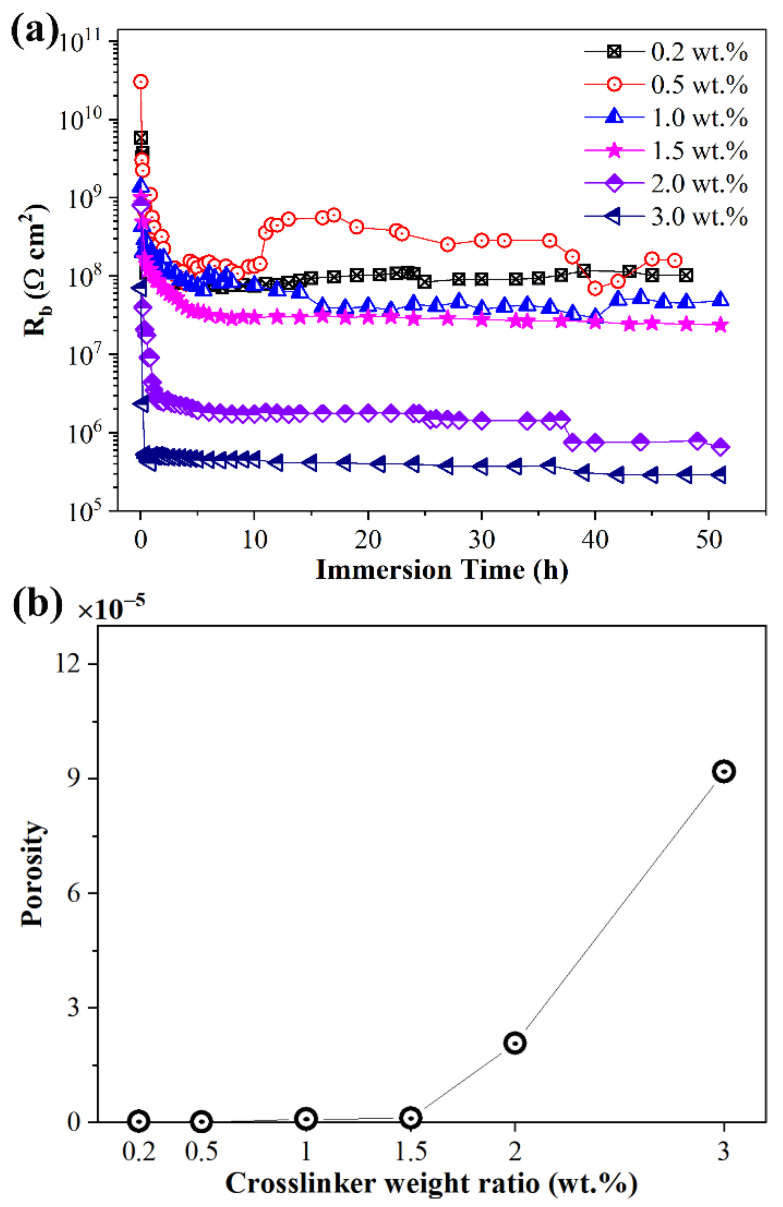
Sample resistance R_b_ as a function of NaCl solution immersion time t (**a**) and porosity (**b**) of silicone rubber sample with diverse crosslinker weight ratio after corona discharge.

**Figure 8 materials-15-06833-f008:**
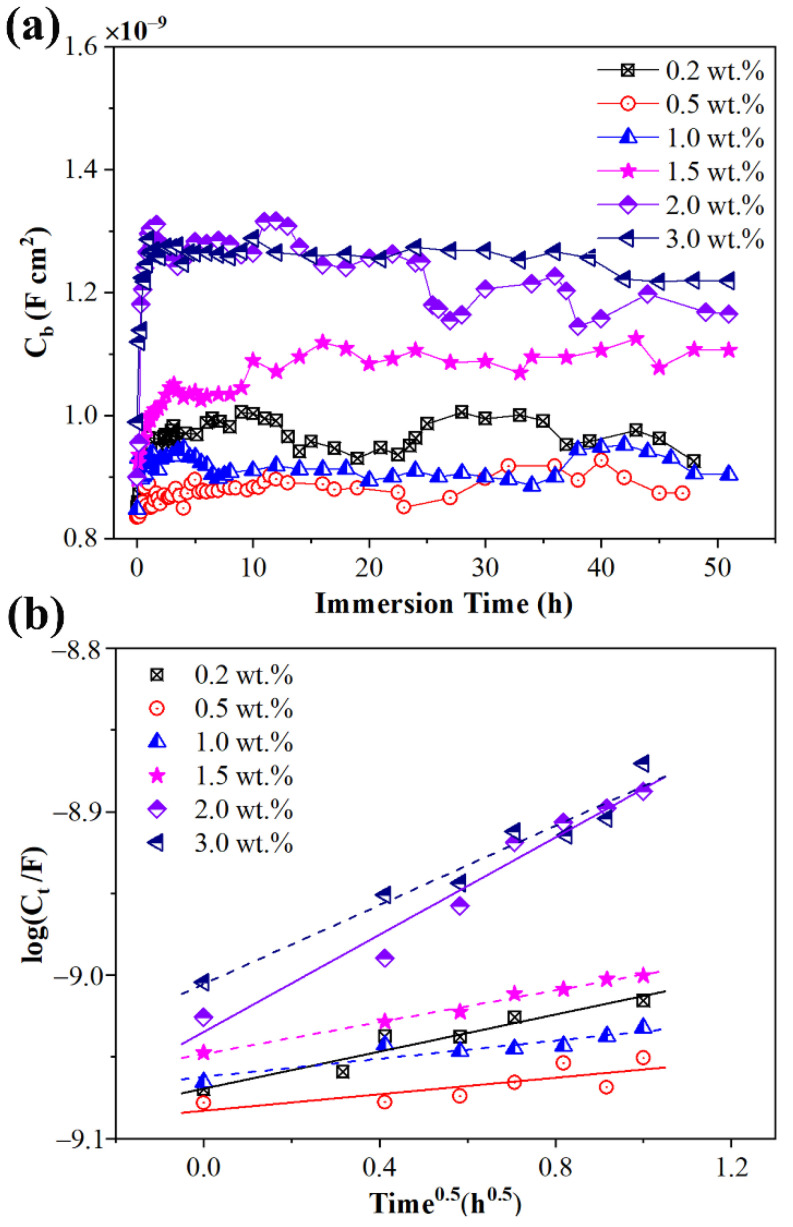
Sample capacitances C_b_ as a function of NaCl solution immersion time *t* (**a**) and logC_b_ as a function of t^1/2^ at initial immersion period (**b**) of samples with diverse cross-linker weight ratio after corona discharge.

**Figure 9 materials-15-06833-f009:**
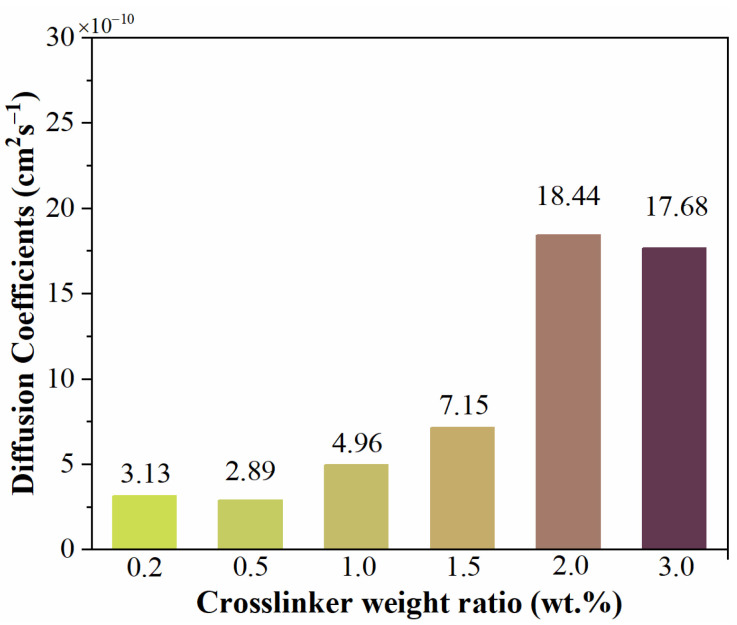
Diffusion coefficients of silicone rubber with diverse crosslinker weight ratio after corona discharge.

**Table 1 materials-15-06833-t001:** Parameters of free volumes in silicone rubber with diverse weight ratios of crosslinkers.

Weight Ratio (wt.%)	τ_3_ (ns)	I_3_ (%)	R_f_ (Å)	V_f_ (Å^3^)
0.2	3.45 ± 0.03	12.1 ± 0.3	3.92	252
0.5	3.40 ± 0.03	12.0 ± 0.3	3.89	246
1.0	3.37 ± 0.03	11.8 ± 0.2	3.87	242
1.5	3.30 ± 0.03	12.1 ± 0.2	3.82	234
2.0	3.28 ± 0.03	12.5 ± 0.2	3.81	232
3.0	3.30 ± 0.03	12.7 ± 0.2	3.83	235

## Data Availability

Raw data is available upon request.

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
