# Peer review of "Influence of Crosslinking Extent on Free Volumes of Silicone Rubber and Water Diffusion after Corona Discharge"

_materials, 2022, doi:10.3390/ma15196833_

Round 1

Reviewer 1 Report

An interesting paper combining the use of several methods to evaluate the surface chemical structure, surface micro morphology and water diffusion of the samples.

Row 88: I don't know the exact publishing system of the journal, but is it possible to refer to the figures (Fig. S1 - S3) that are in the document named "materials-1915912-non-published"?

Concerning the file "materials-1915912-non-published": Below the Figure S3 is wrong value of the breakdown field of air. Instead of 30 kV/mm  it should be 3 kV/mm.

Row 118: I assume the measurement technique was ATR not transmission, please specify. Also include N. of scans, resolution and crystal used.

It is difficult to evaluate the rightness of the results when not specified FTIR technique (ATR, transmission). The intensity and the height of peak depends on force and on the he depth of light penetration in the case of ATR measurement. Also it isn’t specify how many samples were measurement at each crosslinker weight ratio. It looks that only one. If more, it could be included error lines.  

Row 412: probably should be “in high crosslinking level silicone

Please improve especially description of the FTIR measurements. Results are likely to be correct if they correlate with other methods, but this cannot be evaluated in the current form.

Reviewer 2 Report

This paper treats the question of modification induced in rubbers by Corona experiments. The presentation of result is clear, but some parts would ask the style corrections.

Why the authors did not mentioned the name of crosslinker in the anstract?

Why the authors did not comment the stability behavior of materials sublected to the electrical field in the corona experiments?

Why the referenecs are predominantly Chinese?

I suggest some additional comemnts to be added into this manuscript, which would present the omitted the up-mentioned shortcomings.

Reviewer 3 Report

The content of this study is interesting, and the results are useful for the researchers in the related field. However, introduction section should be improved. Especially, sufficient background about  aluminum hydroxide in silicone rubber should be provided. 

Round 2

Reviewer 1 Report

The comments are incorporated and the paper is adequately revised and suitably supplemented. I have no further comments.